# Clinical Application of Intravitreal Aflibercept Injection for Diabetic Macular Edema Comparing Two Loading Regimens

**DOI:** 10.3390/medicina59030558

**Published:** 2023-03-12

**Authors:** Yoo-Ri Chung, Kyung Ho Lee, Kihwang Lee

**Affiliations:** 1Department of Ophthalmology, Ajou University School of Medicine, Suwon 16499, Republic of Korea; 2Love Eye Clinic, Hwaseong 18309, Republic of Korea

**Keywords:** aflibercept, diabetic retinopathy, macular edema, intravitreal injection

## Abstract

*Background and Objectives*: We investigated and compared the efficacy of three and five monthly loading regimens of an intravitreal aflibercept injection (IVA) in patients with diabetic macular edema (DME). *Materials and Methods*: This was a retrospective study that included patients diagnosed with DME and treated with an either three or five monthly aflibercept loading regimen from July 2018 to March 2022. Information on clinical characteristics and changes in the central retinal thickness (CRT) were obtained from medical records. *Results*: In total, 44 eyes of 44 patients with DME treated with IVA were included in this study, with 30 eyes treated with 3-monthly loadings (three-loading group) and 14 eyes with 5-monthly loadings (five-loading group). The mean CRT significantly decreased from the baseline one month after loading in both the three-loading and five-loading groups (*p* < 0.001). Four cases were refractory to treatment in the three-loading group, while there were no cases of refractory DME in the five-loading group. The stability rate was significantly higher in the five-loading group at three months after loading (*p* = 0.033). *Conclusions*: Five-monthly loading regimens of IVA might be favorable for DME considering the rate of refractory cases, stable duration, and the importance of early responsiveness to IVA in DME.

## 1. Introduction

Diabetic macular edema (DME) is one of the main causes of chronic visual impairment and vision loss in the working adult population with diabetic retinopathy (DR) [1]. The prevalence of diabetes mellitus (DM) is increasing worldwide, leading to an increase in the incidence of DR and DME. The International Diabetes Federation estimated the number of individuals with DM to be 463 million in 2019 and projected it to be 700 million by 2045 [2]. In 2020, the numbers of adults with DR and DME were estimated to be 103.12 million and 18.83 million, respectively; by 2045, the numbers are projected to increase to 160.50 million and 28.61 million [3]. Complications of proliferative DR that also cause vision loss in patients with DM include vitreous hemorrhage, tractional retinal detachment, and proliferative vitreoretinopathy [4]. Vascular endothelial growth factor commonly acts on the above complications [4,5,6]. 

Currently, the most widely used treatment for DME is an intravitreal injection of anti-vascular endothelial growth factor (VEGF) [6]. The three anti-VEGF agents ranibizumab, bevacizumab, and aflibercept have proven to be effective in treating DME and improving visual acuity [7,8,9]. Among these anti-VEGF agents, aflibercept (which has a longer acting duration than that of the other anti-VEGF agents) has been approved for the treatment of DME based on the results of several landmark studies [10,11].

However, the study design of the initial loading regimen differed between studies. The VISTA and VIVID studies showed significantly better functional and anatomical outcomes after an aflibercept administration every four or eight weeks after five initial monthly injections than those after laser photocoagulation [12]. In the DA VINCI study, aflibercept was initiated with three monthly doses and then administered as needed (PRN; Pro Re Nata), showing consistent results with an administration of aflibercept every 4 weeks [13]. Furthermore, the one-year results of protocol T showed an improvement of 13.3 letters in the mean visual acuity letter score from the baseline to one year, with six initial monthly doses followed by PRN dosing [7]. Based on these clinical trials and subsequent studies, many experts agree that DME treatment by blocking VEGF typically requires several monthly doses at the beginning to maximize the clinical outcomes over time [14].

There is no consensus on the efficacy of loading doses and the cost-effectiveness of repeat injections due to the lack of studies that directly compare the effects of aflibercept injections after loading. A recent review revealed that the criteria on loading frequency varied by study, ranging from three to six consecutive monthly injections [15]. The national health insurance system in South Korea allowed only three consecutive injections of anti-VEGF agents for DME patients until the year 2019. The five-consecutive loading regimen was covered by the national health insurance system in South Korea starting in the year 2020. The purpose of this study was to compare the two regimens to identify any benefits of five loading injections despite clinical and economic burdens. Accordingly, we conducted this study to identify an effective treatment method by examining the clinical features and results of patients who initially received three or five monthly injections of aflibercept.

## 2. Materials and Methods

The medical records of patients diagnosed with DME and treated with a three- or five-monthly loading regimen of aflibercept (Eylea^®^; Bayer HeathCare, Berlin, Germany) at the Ophthalmology Department of Ajou University Hospital from July 2018 to April 2021 were retrospectively reviewed. This study was approved by the Institutional Review Board of Ajou University Hospital, Suwon, Korea (IRB No.: AJIRB-MED-MDB-21-707) and complied with the Declaration of Helsinki.

DME was identified by optical coherence tomography (OCT) using a Heidelberg SPECTRALIS^®^ OCT device (Heidelberg Engineering, Heidelberg, Germany). The CRT was defined as the distance from the hyperreflective line of the internal limiting membrane to the hyperreflective line of the retinal pigment epithelium (Bruch’s membrane) complex [16] and was obtained using the automatically generated thickness map protocol of the OCT device. Representative cases are presented in the Appendix A. DME was defined as CRT ≥320 μm in men or ≥305 μm in women on OCT [9]. The exclusion criteria were as follows: (1) age < 20 years, (2) macular edema suspected to originate from factors other than DME, (3) a history of pars plana vitrectomy, (4) prior intravitreal anti-VEGF injection within two months, (5) prior steroid injection (intravitreal or posterior sub-Tenon) within six months, (6) focal/grid photocoagulation or panretinal photocoagulation within the previous six months, (7) active intraocular inflammation or infection in either eye, and (8) uncontrolled glaucoma in either eye. Only one eye was randomly selected and enrolled in the current study if a patient received an intravitreal aflibercept injection (IVA) in both eyes.

Medical history, clinical characteristics, and information regarding current diabetic medications were obtained from individuals’ medical records. Blood pressure was measured during each injection visit (including both systolic and diastolic values). Glycated hemoglobin (HbA1c) data were collected from the preceding three months prior to the first IVA, and the DR grade was assessed using fundus photographs and fluorescein angiography findings. IVA (2 mg/0.05 mL) was administered in a standard manner by one of the three participating retinal specialists. A baseline OCT was performed in the week before the initial loading IVA and was repeated one month after the last loading IVA. Refractory DME was defined as a CRT decrease <10%, the occurrence of new subretinal fluid (SRF) and/or intraretinal fluid (IRF), or a lesion of SRF or IFR found after loading injections. As the number of loading injections differed by groups, the follow-up months were counted from the last loading injection (Figure 1). 

All patients underwent ophthalmic examinations at every monthly visit, including a slit lamp examination, dilated fundus examination, intraocular pressure measurement, and OCT. An additional treatment was performed if necessary according to the retreatment criteria of (1) CRT at one month after IVA loading ≥320 μm (male) or ≥305 μm (female); (2) if CRT increased by more than 10% compared to CRT at one month after IVA loading; and (3) new SRF or IRF. If the OCT findings did not meet the retreatment criteria, they were defined as “stable”. The stability rate was defined as the proportion of “stable” eyes at each time point. Additional treatments included aflibercept and other anti-VEGF agents, the administration of steroids, focal lasers, or vitrectomy according to the clinical judgement of each clinician. 

All statistical analyses were performed using the Statistical Package for the Social Sciences (SPSS) version 22.0 (IBM SPSS, IBM Corp., Armonk, NY, USA). The chi-square test and independent *t*-test were used to compare categorical and continuous variables, respectively. A paired *t*-test was used to detect changes in numerical values at each time point of the study relative to the baseline values within the groups. A repeated-measures analysis of variance (ANOVA) test was performed to verify differences in the changes in CRT between the 3-loading and 5-loading groups. Logistic regression analysis was performed to identify factors associated with a stable status following loading injections: this is presented as the odds ratio with a 95% confidence interval and *p* value. Statistical significance was set at a *p* value < 0.05.

## 3. Results

In total, 44 eyes from 44 patients with DME were included in this study. Among these eyes, 30 were treated with 3 IVA loading regimens and 14 were treated with 5 IVA loading regimens. The mean age was 57.7 ± 11.8 years (range: 39–85). The demographic characteristics of the patients are summarized in Table 1. The use of dipeptidyl peptidase-4 (DPP-4) inhibitors was more frequent among antidiabetic medications in the three-loading group, but there were no significant differences in the baseline characteristics, such as the duration of diabetes or HbA1c level.

There was no statistically significant difference in initial CRT between the two groups (*p* = 0.437). The mean CRT decreased from 490.8 ± 123.5 μm at the baseline to 308.2 ± 89.1 μm at one month after loading in the three-loading group (*p* < 0.001), whereas it decreased from 461.4 ± 95.1 μm at the baseline to 320.9 ± 65.2 μm at one month after loading in the five-loading group (*p* < 0.001). Four cases were resistant to treatment in the three-loading group, while there were no resistant cases in the five-loading group. The stability rate was significantly higher in the five-loading group at three months after loading (*p* = 0.033); however, there was no significant difference detected thereafter (Figure 2, Table 2).

Time to recurrence was longer in the five-loading group compared to that in the three-loading group, although the difference was not significant (152.2 ± 62.6 vs. 116.6 ± 64.6 days, *p* = 0.156). There was a statistically significant difference in the total number of IVAs administered (including initial loading between the two groups) (4.2 ± 1.5 in three-loading vs. 6.1 ± 0.9 five-loading; *p* < 0.001), but there was no difference in the number of additional IVAs administered during the 6 months after the initial loading therapy. 

Logistic regression analysis indicated that the CRT reduction rate was a significant factor for stabilization for three months after loading (odd ratio 1.067, 95% confidence interval 1.010–1.128, *p* = 0.021), while there were no significant variables for six months stabilization (Table 3). There were no specific antidiabetic medications associated with the stabilization of DME.

## 4. Discussion

DME is a chronic and sight-threatening disease that significantly impacts quality of life [17,18]. Microvascular abnormalities and occlusions due to chronic hyperglycemia are considered the main etiologies of DME, which is associated with various growth factors. Among these growth factors, VEGF causes an abnormal occlusive function of the inner blood–retinal barrier and causes an accumulation of extracellular interstitial fluid [19,20]. Since anti-VEGF agents were introduced based on the mechanisms of DME, intravitreal injections of anti-VEGF agents have replaced laser photocoagulation as the standard treatment for most DME patients [6]. Aflibercept is a widely used anti-VEGF agent for DME and is active longer after administration than other such drugs [10].

In this study, both the three and five initial loading regimens of aflibercept were effective for the treatment of DME. The non-recurrence rate at three months after loading was significantly higher in the five-loading group than that in the three-loading group. Although not statistically significant, this tendency persisted for six months. The difference between the two groups decreased with time and was almost the same at six months after loading. This trend suggests that an initial five-loading might be more effective than three-loading in the short term, but the effect gradually decreases over time in both groups. Six months after the loading treatment, an additional treatment was required in approximately two-thirds of the patients in both groups. It is believed that a larger number of loading injections might lead to a smaller number of refractory DME patients and longer stable periods without the recurrence of DME, which is important for optimal improvement in visual acuity through an early intervention. Moreover, large fluctuations in macular thickness are associated with poorer visual outcomes in eyes with DME treated with anti-VEGF injections [21,22]. Reducing the variability in DME based on macular thickness is a benefit of a five-loading regimen and can result in better visual outcomes in the long term.

Although shown to be effective for DME, IVA does not evoke a response in all cases. A post hoc analysis of Protocol T evaluated the proportion of eyes with persistent DME after 24 weeks of treatment with aflibercept, bevacizumab, or ranibizumab [23]. Persistent DME for 24 weeks was less likely with aflibercept than with bevacizumab or ranibizumab; however, 31.6% of eyes did not show an adequate response to aflibercept [23]. In this study, we found four refractory cases of DME, all of which were in the three-loading group. We cannot be 100% sure that these refractory cases might have responded if treated with a five-loading regimen; however, the absence of refractory cases in the five-loading group is worth considering when determining the proper loading regimen for DME. Aflibercept showed good effects when replacing bevacizumab or ranibizumab in refractory DME [24]. A recent Diabetic Retinopathy Clinical Research (DRCR) study reported no significant differences in 2-year visual outcome for DME between aflibercept monotherapy and bevacizumab replaced with aflibercept, suggesting a rescue effect of later applied IVA [25]. However, in cases that do not respond to IVA, alternative methods such as steroids, laser treatment, or even surgery may be needed to treat the DME [15,26]. Of the four refractory cases presented in our study, two were treated with focal laser photocoagulation, while the others received intravitreal corticosteroid injections. In one patient treated with steroids, vitrectomy was performed for resistant DME.

The prevalence of DME is increasing, highlighting the need for long-term treatment and placing significant burden on patients and insurance costs. DME is not only a chronic disease but it also occurs in both eyes in most cases; therefore, it is essential to find an effective and economical treatment regimen. There are many variations in initial treatment schedules, including 3 to 6 monthly consecutive anti-VEGF injections [15]. Furthermore, many strategies were studied to maintain and maximize the effect of anti-VEGF treatment after the loading period. A fixed bimonthly or a pro-re-nata regimen (based on strict monitoring and retreatment criteria, as stated in the DRCR.net protocol T) is recommended [14]. Treat-and-extend therapy is also a recommended regimen with non-inferior visual and anatomical improvement in DME compared to fixed dosing regimens [27,28], while less visual improvement was noted when the longest treatment interval was 16 weeks [29]. Despite these efforts, DME may persist. For such cases, there has been a report of meaningful gains in vision (with little risk of vision loss) with a continuous IVA treatment [23]. An IVA regimen of at least six consecutive injections showed efficacy in 50% of non-responders to bevacizumab [30,31]. However, a continuous IVA treatment (such as that in phase III trials) is not always possible in clinical practice [32,33]. In the Korean National Health Insurance system, the number of anti-VEGF treatments covered is limited per patient, and the maximal effect should be obtained with limited resources. As shown in our study, a focus on early loading IVA might be beneficial in real-world practice, where a frequent injection such as with fixed doses in clinical trials is often limited. A five-loading regimen will reduce CRT fluctuations in patients with DME, which was reported to be associated with better visual outcomes based on a post hoc analysis from the DRCR Network protocols T and V clinical studies [21].

The effects of DPP-4 inhibitors on DME rarely have been investigated, although one study revealed no such influence [34]. In our study, DPP-4 inhibitors were used more frequently in the three-loading group compared to the five-loading group. However, their association with the stability of DME was not evident, which might be due to the similar glucose controls in the two groups despite different proportions of antidiabetic medications.

This study has several limitations related to its retrospective nature, the small number of included patients, and the relatively short follow-up period. As mentioned above, a longer follow-up period might demonstrate a different tendency. The difference in the numbers of patients in the groups is also a limitation and was affected by the change in the coverage policy of the national health insurance system in South Korea. In addition, no untreated DME group was included in this retrospective study. Moreover, the treatment response to IVA in DME is limited to anatomical improvement. Further prospective, randomized, large-scale, well-controlled trials may provide evidence for the optimal loading IVA treatment method for DME.

## 5. Conclusions

Despite the limitations, this study provides a valuable insight into the optimal loading treatment regimen of IVA for DME. We suggest applying a five-loading regimen in eyes with DME based on the rate of refractory cases, stable duration, and the importance of early responsiveness to IVA in DME.

## Figures and Tables

**Figure 1 medicina-59-00558-f001:**
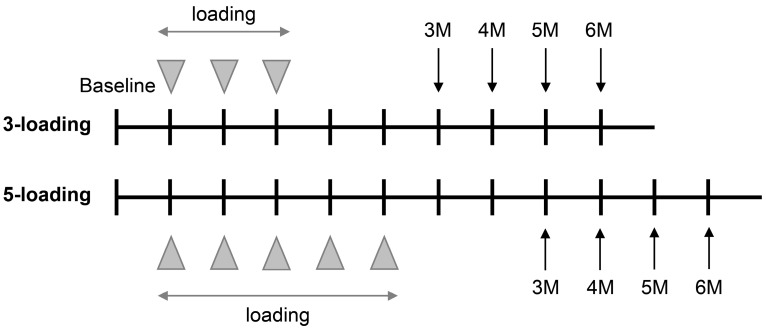
Scheme of treatment and follow-up protocol by intravitreal aflibercept loading regimen in eyes with diabetic macular edema. Follow-up period is presented as months (M) after the last loading injection.

**Figure 2 medicina-59-00558-f002:**
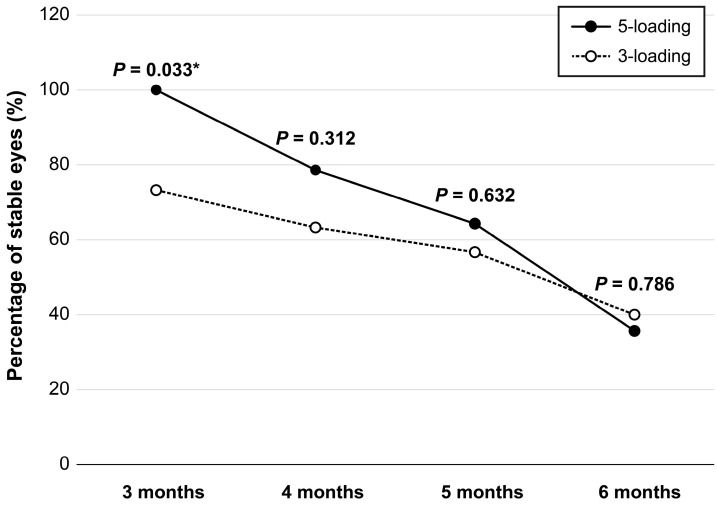
Proportion of stable eyes with DME following IVA loading. Follow-up period is presented as months after the last loading injection. * *p* < 0.05 by chi-square test.

**Table 1 medicina-59-00558-t001:** Baseline characteristics of included patients with diabetic macular edema.

Variables	3 Injections	5 Injections	*p* Value
No. of eyes	30	14	
Age (years)	57.8 ± 12.0	57.6 ± 11.7	0.960 †
Sex, male	18 (60.0%)	7 (50.0%)	0.533 *
Treatment—naïve	9 (30.0%)	2 (14.3%)	0.262 *
Hypertension	14 (46.7%)	8 (57.1%)	0.517 *
Systolic BP (mmHg)	138.7 ± 19.4	134.9 ± 16.3	0.525 †
Diastolic BP (mmHg)	75.5 ± 16.3	79.4 ± 12.8	0.330 †
Antidiabetic medications			
Biguanides	22 (73.3%)	9 (64.3%)	0.428 *
Sulfonylurea	20 (66.7%)	9 (64.3%)	0.759 *
SGLT-2 inhibitor	1 (3.3%)	3 (21.4%)	0.094 *
DPP-4 inhibitor	20 (66.7%)	5 (35.7%)	0.038 *
Thiazolidinedione	2 (6.7%)	4 (28.6%)	0.055 *
Insulin	7 (23.3%)	4 (28.6%)	0.755 *
DM duration (years)	14.4 ± 9.8	12.1 ± 7.5	0.449 †
NPDR	14 (46.7%)	9 (64.3%)	0.495 *
Chronic kidney disease	7 (23.3%)	0	0.078 *
HbA1c (%)	7.3 ± 1.3	7.7 ± 1.5	0.422 †

BP = blood pressure, DM = diabetes mellitus, DPP-4 = dipeptidyl peptidase-4, NPDR = non-proliferative diabetic retinopathy, SGLT-2 = sodium glucose cotransporter-2. Data are expressed as numbers (percentages) for categorical values and means ± standard deviations for numeric values. * Chi-square test. † Independent *t*-test.

**Table 2 medicina-59-00558-t002:** Clinical outcomes through 6 months of follow-up.

**Variables**	**3 Injections**	**5 Injections**	***p* Value**
No. of eyes	30	14	
Baseline CRT (μm)	490.8 ± 123.5	461.4 ± 95.1	0.437 *
CRT at 1 month after loading (μm)	308.2 ± 89.1	320.9 ± 65.2	0.635 *
*p* value within group	<0.001 ^†^	<0.001 ^†^	0.350 ^‡^
CRT change from baseline (μm)	−182.6 ± 151.4	−140.5 ± 99.6	0.350 *
CRT change from baseline (%)	−34.0 ± 21.6	−28.7 ± 15.4	0.832 *
Refractory DME	4 (13.3%)	0	0.152 ^§^
No. of focal laser	0.3 ± 0.5	0.2 ± 0.4	0.674 *
Stable for 3 months	22 (73.3%)	14 (100.0%)	0.033 ^§^
Stable for 4 months	19 (63.3%)	11 (78.6%)	0.312 ^§^
Stable for 5 months	17 (56.7%)	9 (64.3%)	0.632 ^§^
Stable for 6 months	12 (40.0%)	5 (35.7%)	0.786 ^§^
Time to recurrence (days)	116.6 ± 64.6	152.2 ± 62.6	0.153 *
No. of IVA (including loading)	4.2 ± 1.5	6.1 ± 0.9	<0.001 *
No. of additional IVA after initial loading	1.2 ± 1.5	1.1 ± 0.9	0.761 *

CRT = central retinal thickness, DME = diabetic macular edema, IVA = intravitreal aflibercept injection. Data are expressed as numbers (percentages) for categorical values and means ± standard deviations for numeric values. * Independent *t*-test. ^†^ Paired t-test. ^‡^ Repeated measures ANOVA; ^§^ Chi-square test.

**Table 3 medicina-59-00558-t003:** Factors associated with stability of eyes for 3 months vs. 6 months after loading.

	For 3 Months after Loading			For 6 Months after Loading		
	OR	95% CI	*p* Value	OR	95% CI	*p* Value
Age	1.043	0.966–1.125	0.282	0.972	0.919–1.027	0.304
Sex, male	1.400	0.301–6.505	0.668	1.702	0.488–5.934	0.404
Loading, 5 times			0.999	0.833	0.224–3.103	0.786
Treatment—naïve	1.000	0.170–5.866	1.000	2.400	0.598–9.637	0.217
Baseline CRT	0.998	0.992–1.004	0.539	0.994	0.987–1.000	0.062
Change (%) of CRT after loading	1.067	1.010–1.128	0.021 *	1.015	0.983–1.047	0.370
Hypertension	3.750	0.665–21.154	0.134	1.786	0.523–6.100	0.355
Chronic kidney disease	1.400	0.144–13.568	0.772	2.476	0.476–12.716	0.282
Biguanides	1.040	0.173–6.258	0.966	0.844	0.228–3.432	0.884
Sulfonylurea	1.705	0.325–8.933	0.528	0.815	0.223–2.982	0.757
SGLT-2 inhibitor			0.999	5.357	0.508–56.502	0.163
DPP-4 inhibitor	0.500	0.085–2.926	0.442	0.471	0.135–1.641	0.237
Thiazolidinedione			0.999	0.733	0.119–4.525	0.738
Insulin	0.381	0.070–2.066	0.263	0.835	0.203–3.444	0.803
DM duration	1.039	0.950–1.137	0.402	1.004	0.938–1.074	0.906
PDR	0.714	0.154–3.319	0.668	1.912	0.558–6.554	0.302
HbA1c (%)	1.835	0.809–4.162	0.146	1.121	0.682–1.840	0.653

* *p* value < 0.05 by logistic regression analysis. CRT = central retinal thickness, DM = diabetes mellitus, DPP-4 = dipeptidyl peptidase-4, PDR = proliferative diabetic retinopathy, SGLT-2 = sodium glucose cotransporter-2.

## Data Availability

The data presented in this study are available on request form the corresponding author.

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
