# Peer review of "Clinical Application of Intravitreal Aflibercept Injection for Diabetic Macular Edema Comparing Two Loading Regimens"

_medicina, 2023, doi:10.3390/medicina59030558_

Round 1

Reviewer 1 Report

It is not apparent to me why the authors chose to treat some patients with 3 loading doses and others with 5 in this study? Was it a policy change at their institute? This is important to further understand the data being analysed. For example, some eyes may have been treated with 5 loading doses just cause they did not adequately respond to 3 injections. I understand this is a retrospective study but why was DME management at your institute variable?

The duration of the study and time points of analysis are a major limitation. It would be interesting to see on the long run whether 3 vs 5 loading doses had an effect on long term vision. This should be interpreted in the light of the recent DRCR study comparing early aflibercept vs late aflibercept after bevacizumab first (PMID: 35833805). Delayed treatment with aflibercept did not affect final vision in this study which is relevant to yours.

please discuss your findings in light of this recent review (PMID: 36436614)

Author Response

It is not apparent to me why the authors chose to treat some patients with 3 loading doses and others with 5 in this study? Was it a policy change at their institute? This is important to further understand the data being analysed. For example, some eyes may have been treated with 5 loading doses just cause they did not adequately respond to 3 injections. I understand this is a retrospective study but why was DME management at your Institute variable?

>> Thank you for your comment. There is a lack of consensus on adequate treatment schedule of injections for DME. The national insurance system in South Korea allowed only 3 consecutive anti-VEGF injections of aflibercept for DME patients until year 2019. The five-consecutive loading regimen was covered by national health insurance system in South Korea starting in year 2020. Thus, some patients were treated with the maximum 5 consecutive injections. This also explains why the sample size of the 3-loading group is more than twice that of the other group. The purpose of this study is to compare the two regimens to identify any benefit of 5 loading injections despite clinical and economic burden. We agree that this point needed to be explained, and we added this in the Introduction (lines 53-60, page 2).

The duration of the study and time points of analysis are a major limitation. It would be interesting to see on the long run whether 3 vs 5 loading doses had an effect on long term vision. This should be interpreted in the light of the recent DRCR study comparing early aflibercept vs late aflibercept after bevacizumab first (PMID: 35833805). Delayed treatment with aflibercept did not affect final vision in this study which is relevant to yours.

>> Thank you for your comment; we agree that the short-term follow-up is one of the major limitations of this study. The randomized controlled study by Jhaveri et al. (PMID 35833805) compared visual outcomes in patients receiving aflibercept monotherapy to those treated initially with bevacizumab and switched to aflibercept therapy. They revealed no significant differences in visual outcomes over a 2-year period. We added this at lines 208-210, page 7.

The review that you recommended below (PMID 36236614) revealed that the difference of visual gain by type of anti-VEGF agents was not significant at longer follow-up. This also suggests the possibility of different outcome results when a longer follow-up period is applied.   

Please discuss your findings in light of this recent review (PMID: 36436614).

>> Thank you for suggesting the recent review on persistent diabetic macular edema (DME) by Sorour et al. (PMID 36436614). The review has discussed various methods and criteria regarding loading frequency and pathogenic factors associated with responsiveness. We included such information in the Introduction (lines 53-54, page 2). We also cited a reference defining refractory DME (lines 194-197, page 7).

Reviewer 2 Report

Material and Methods: It would be worthwhile for authors to show the measurement of CRT with representative images for both groups, to confirm that images were acquired from approximately the same location (macula).

Second paragraph: was post-injection OCT performed at one month or ‘starting’ one month after IVA? Please check abbreviations eg IRF and IFR.

Results: Since authors had access to DR grade of patients, did the loading regimens affect DR grade post loading? Were there other aspects of DR that was improved or worsened by either regimen?

Table 2 title should be corrected.

Table 3 title: ‘by regimen’ may read better than ‘by period’.

Discussion: Needs to be improved with more references (especially paragraph 2). There was a significant difference in the proportion of patients with DPP-4 inhibitor (Table 1). Authors must clearly describe this in the results and discuss how that impacts their findings. Also, authors might want to double-check English language in paragraph 1.

Paragraph 2: Authors claim that both regimens were effective for treating DME. Without placebo / untreated controls, this claim is inaccurate as there is no comparison with patients who would undergo spontaneous or untreated regression.

Differences in stable eyes was only significant 3 months post-treatment. Authors should discuss the clinical significance of this short-term improvement and whether it is worth it given patients have to undergo 2 additional invasive injections.

Limitations: Authors discussion of limitations must be improved. For example, how the huge difference in sample size (sample size of 3-loading group is more than twice that of the other group) affects the interpretation of the data should be made clear. Also, limitations imposed by the absence of untreated / placebo group should be discussed.

Author Response

Material and Methods: It would be worthwhile for authors to show the measurement of CRT with representative images for both groups, to confirm that images were acquired from approximately the same location (macula).

>> Thank you for your comment. As mentioned in the Methods, CRT was measured automatically by OCT (Heidelberg SPECTRALIS® OCT device, Heidelberg Engineering, Heidelberg, Germany). We added a supplementary figure of representative images for both groups (Figure S1).

Results: Since authors had access to DR grade of patients, did the loading regimens affect DR grade post loading? Were there other aspects of DR that was improved or worsened by either regimen?

>> As the follow-up period was relatively short in this study, the change of DR grade following IVA was not investigated.

Table 2 title should be corrected.

>> We corrected the title of Table 2 as follows: “Clinical outcomes through 6 months of follow-up.”

Table 3 title: ‘by regimen’ may read better than ‘by period’.

>> Table 3 presents factors associated with stability at 3 months of follow-up (at left column), i.e., immediate response, and at 6 months of follow-up (at right column). The loading regimen was one of the investigated variables, and ‘by period’ signified 3 months or 6 months of follow-up. We revised the title of Table 3 as follows: “Factors associated with stability of eyes for 3 months vs. 6 months after loading.”

Discussion: Needs to be improved with more references (especially paragraph 2). There was a significant difference in the proportion of patients with DPP-4 inhibitor (Table 1). Authors must clearly describe this in the results and discuss how that impacts their findings. Also, authors might want to double-check English language in paragraph 1.

>> Thank you for your comment. The proportion of DPP-4 inhibitors was significantly higher in the 3-loading group, while use of DPP-4 inhibitors was not significantly associated with stabilization of DME (Table 3). Although not statistically significant, the use of SGLT-2 inhibitors and thiazolidinediones was more frequent in the 5-loading group. We think that the lack of association of DME and DPP-4 inhibitors may be due to the similar diabetic control in the two groups, as presented by HbA1c level. The glycemic control is a well-known factor associated with DR complications, while the effects of antidiabetic medications remain uncertain. We added a brief discussion on DPP-4 inhibitors in the Discussion (pages 11-12).

We re-edited the whole manuscript, including the paragraph recommended.

Paragraph 2: Authors claim that both regimens were effective for treating DME. Without placebo / untreated controls, this claim is inaccurate as there is no comparison with patients who would undergo spontaneous or untreated regression.

>> The definitions of anatomical responses may vary by study. Although we did not compare the results to a placebo / untreated group, CRT reductions were evident following IVA in both 3-loading and 5-loading groups (P < 0.001 by paired t-test, Table 2). We defined refractory DME as a CRT decrease < 10%, so an anatomical response implied a CRT decrease ≥ 10%. In this aspect, we believe that both regimens showed effectiveness for DME improvement.  

Differences in stable eyes was only significant 3 months post-treatment. Authors should discuss the clinical significance of this short-term improvement and whether it is worth it given patients have to undergo 2 additional invasive injections.

>> Thank you for your comment. We think that the early responsiveness shown in the 5-loading group may have clinical significance in terms of long-term visual outcomes in eyes with DME. A number of papers have shown that large fluctuations in macular thickness are associated with poor visual outcomes in eyes with DME treated with anti-VEGF injections. Reduced variability in DME through change in macular thickness can be a benefit of the 5-loading regimen, which demonstrated better visual outcomes in the long term. We added this in the Discussion (page 9-10), citing the following references:

Wang VY, et al. Fluctuations in macular thickness in patients with diabetic macular oedema treated with anti-vascular endothelial growth factor agents. Eye 2022;36:1461-4167 (PMID 34234291)

Starr MR et al. Fluctuations in central subfield thickness associated with worse visual outcomes in patients with diabetic macular edema in clinical trial setting. Am J Ophthalmol 2021;232:90-97 (PMID 34283986)

Limitations: Authors discussion of limitations must be improved. For example, how the huge difference in sample size (sample size of 3-loading group is more than twice that of the other group) affects the interpretation of the data should be made clear. Also, limitations imposed by the absence of untreated / placebo group should be discussed.

>> Thank you for your comment. We addressed this in the limitations (lines 248-249, page 8).

Reviewer 3 Report

authors wrote an interesting article 

I recommend the following improvements:

1) please expand introduction. I'd add a short sentence regarding long term complication of diabetes. Please have a look and add the following paper: PMID: 24227910 for PVR.

2) please expand the study-limitations and add future possible studies 

Author Response

I recommend the following improvements:

1) Please expand introduction. I'd add a short sentence regarding long term complication of diabetes. Please have a look and add the following paper: PMID: 24227910 for PVR.

>> The article that the reviewer mentioned focused on the genetic pathogenesis of PVR associated with VEGF-A, Otx homeobox, and p53 family genes. We added briefly complications of diabetic retinopathy such as vitreous hemorrhage, tractional retinal detachment, and PVR at lines 33-36.  

2) Please expand the study-limitations and add future possible studies

>> This was a common point of the reviewers, so that we added the limitations of this study at lines 248-255.  

Round 2

Reviewer 2 Report

The authors have made sufficient revisions to improve the manuscript to merit consideration for publication.

Reviewer 3 Report

approved